



# Characterizing Uncertainties of Earth System Modeling with Heterogeneous Many-core Architecture Computing

Yangyang Yu[1,2], Shaoqing Zhang[*1,2,3], Haohuan Fu[*4,5], Lixin Wu[*1,2,3], Dexun Chen[*5], Yang Gao[3,6], Zhiqiang Wei[3], Dongning Jia[3], Xiaopei Lin[1,2,3]

[1] Key Laboratory of Physical Oceanography, Ministry of Education, Institute for Advanced Ocean Study, Frontiers Science Center for Deep Ocean Multispheres and Earth System (DOMES), Ocean University of China, Qingdao, 266100, China
[2] College of Oceanic and Atmospheric Sciences, Ocean University of China, Qingdao, 266100, China
[3] Pilot National Laboratory for Marine Science and Technology, Qingdao, 266100, China
[4] Ministry of Education Key Lab. for Earth System Modeling, and Department of Earth System Science, Tsinghua University, Beijing, 100084, China
[5] National Supercomputing Center in Wuxi, Wuxi, 214072, China
[6] Key Laboratory of Marine Environmental Science and Ecology, Ministry of Education, Frontiers Science Center for Deep Ocean Multispheres and Earth System (DOMES), Ocean University of China, Qingdao, 266100, China

*Correspondence to*: Shaoqing Zhang (szhang@ouc.edu.cn), Haohuan Fu (haohuan@tsinghua.edu.cn), Lixin Wu (lxwu@ouc.edu.cn), Dexun Chen (adch@263.net)

**Abstract.** Physical and heat limits of the semiconductor technology require the adaptation of heterogeneous architectures in supercomputers, such as graphics processing units (GPUs) with many-core accelerators and many-core processors with management and computing cores, to maintain a continuous increase of computing performance. The transition from homogeneous multi-core architectures to heterogeneous many-core architectures can produce "potential differences" that lead to numerical perturbations and uncertainties in simulation results, which could blend with errors due to coding bugs. The development of a methodology to identify the computational perturbations and secure the model correctness is a critically important step in model development on the computer system with new architectures. We have developed a methodology to characterize the uncertainties in the heterogeneous many-core computing environment, which contains a simple multiple-column atmospheric model consisting of typical discontinuous physical parameterizations defined by on-off switches, an efficient ensemble-based test approach, and a software tool applied to the GPU-based high-performance computing (HPC) and Sunway systems. Statistical distributions from ensembles of the heterogeneous systems show quantitative analyses of computational perturbations and acceptable error tolerances. The methodology explores fully understanding to distinguish between perturbations caused by platforms and discrepancies caused by software bugs, and provides encouraging references for verifying the reliability of supercomputing platforms and discussing the sensibility of Earth system modeling to the adaptation of new heterogeneous many-core architectures.





# 1 Introduction

The development of numerical simulations requires the increase in computing power. Due to physical and heat limits, regular increases in the number of supercomputing processors came to a stop roughly one decade ago. Since then, there is the transition of supercomputers from homogeneous multi-core to heterogeneous many-core architectures in order to continue

increasing the performance, leading to an environment with multiple types of computing devices and cores. The major computing power of heterogeneous many-core architectures is provided by many-core accelerators such as NVIDIA graphics processing units (GPUs) (Vazhkudai et al., 2018) as well as Intel Xeon Phi MICs (Liao et al., 2014) and many-core processors Sunway computing processing elements (CPEs) (Fu et al., 2016). Heterogeneous many-core architecture computing can produce nonidentical floating-point arithmetic outputs. The differences between arithmetic units and

compilation flows can sometimes cause numerical perturbations and generate uncertainties (Zhang et al., 2020).

Earth system models (ESMs) are based on mathematical equations, including dynamical and parameterization processes, established by dynamical, physical, chemical, and biological processes to resolve more details of interacting atmosphere, ocean, sea-ice, and land surface components through numerical methods consisting of millions of lines of legacy codes (Flato, 2011), such as the Community Earth System Model (CESM). Perturbations can cause sudden changes in

discontinuous physical parameterizations (Yano, 2016) defined by on-off switch structures in programming, such as cloud bottom and cloud top (Zhang and Mcfarlane, 1995) as well as top of planetary boundary layer (Sun and Ogura, 1980) in atmosphere modules and mixed layer depth in ocean modules (Kara et al., 2000).

The traditional method to secure the correctness of ESMs for computing environment changes has been a cumbersome process. For example, data from a climate simulation of several hundred years (typically 400) on the new machine is

analyzed and compared to data from the same simulation on a "trusted" machine by senior climate scientists (Baker et al., 2015). Then, CESM ensemble-based consistency test (CESM-ECT) is currently used to evaluate climate consistency for the ongoing state of computing environment changes (Baker et al., 2015; Milroy et al., 2016). However, all the above-mentioned methods focus on homogeneous multi-core architecture computing. For heterogeneous many-core architecture computing, the difference in computing environments between master and slave cores can cause perturbations whenever a slave core or

an accelerator is involved. There is a lack of methodology for identifying and characterizing the computational perturbations in heterogeneous many-core computing environments.

The goal of this article is to design a methodology to characterize the uncertainties of Earth system modeling in heterogeneous many-core computing environments and discuss its influence on numerical simulations. The methodology contains a simple multiple-column atmospheric model consisting of typical discontinuous physical parameterizations defined

by on-off switches to study uncertainties through sudden changes, an efficient ensemble-based test approach to characterize uncertainties through quantitative analyses, and a software tool to verify the reliability of heterogeneous many-core systems.



The rest of the paper is organized as follows. Section 2 shows background information on uncertainties in floating-point computation. Section 3 details the methodology to characterize uncertainties. Section 4 shows the results of experiments with the methodology. Finally, the summary and discussion are given in Section 5.

## 2 Uncertainties of floating-point computation

### 2.1 The origins of uncertainties

During ESM code porting, changes in computing environments can cause simulation results that are no longer bit-for-bit (BFB) identical to previous output data (Baker et al., 2015). For immutable codes with homogeneous multi-core architecture computing, changes in software and hardware environments such as compilers and instruction sets are the main reason for non-BFB reproducibility (Rosinski and Williamson, 1997). We define floating-point output differences generated by compiler and instruction set changes as "potential differences". **Figure 1** shows the schematic illustration of potential difference sources. In the process of translating high-level programming language into machine language codes, different compilers and/or instruction sets can cause assembly code differences as different code execution order and/or different intermediate register floating-point precision, eventually causing nonidentical floating-point outputs. Generally for homogeneous computing, changing compilers, for example, from Intel to GNU, or changing instruction sets, for example, from SSE to x87 can cause potential differences and generate uncertainties.

### 2.2 Uncertainties in heterogeneous many-core architecture computing

Heterogeneous many-core architectures have to work with their own instruction sets and corresponding compiler adaption. To achieve functional differentiation of master and slave cores, the computing on master and slave cores is generally with different instruction sets and compilers. Therefore, compared with homogeneous computing using the master cores only, heterogeneous computing can cause potential differences whenever a slave core or accelerator is involved. That says that the model is perturbed constantly during integration on a heterogeneous supercomputing platform (Zhang et al., 2020).

For GPU-based HPC systems, GPU devices are introduced as accelerators for general-purpose scientific and engineering applications (Xiao et al., 2013), such as the GPU-based Princeton Ocean Model (POM) (Xu et al., 2015) and the GPU-based COSMO regional weather model by MeteoSwiss (Fuhrer et al., 2018). The fixed instruction set, SASS, and compiler, NVCC, are used in GPU to achieve cost-effective data processing (Stephenson et al., 2015). For the Sunway TaihuLight which is the Chinese homegrown supercomputing platform, master and slave cores are integrated into the same processor, SW26010, as shown in **Fig. 2**. Each SW26010 processor can be divided into four identical core groups (CGs), which are connected through the network on chip. Each CG includes one management processing element (MPE), one CPE cluster with 8x8 CPEs. For the Sunway system, to achieve the maximum aggregated computing power and minimize the complexity of the micro-architecture, the MPEs and CPEs are with different functions so that programs are generally computed in a hybrid mode to use instruction sets separately (Fu et al., 2016). The Sunway TaihuLight has realized high-resolution scientific



computing with high-efficiency, such as Community Atmosphere Model version 5 (CAM5) (Fu et al., 2017a; Fu et al., 2017b) and CESM1.3 (Zhang et al., 2020). Upgraded from SW26010, the new generation Sunway Supercomputer has been

constructed using SW26010P. Using the new generation Sunway Supercomputer, higher-resolution ESMs have been developed. Identifying and understanding the characteristics of floating-point computation uncertainties in heterogeneous architectures are urgently demanded.

To visually illustrate perturbations caused by different computing environments, we start from the Goff-Gratch equation (Goff and Gratch, 1946) and see the floating-point results. The Goff-Gratch equation is a formula that calculates saturated

vapor pressure (SVP), highly-nonlinear and widely used in cloud parameterizations such as Zhang and McFarlane cumulus convection parameterization scheme (ZM scheme) (Zhang and Mcfarlane, 1995) and Morrison and Gettelman double-moment stratiform microphysics scheme (Morrison and Gettelman, 2008). The Goff-Gratch equation is given by Eq.(1):

$$\log p = -7.90298\,(T_{bt}/T - 1) + 5.02808\,\log(T_{bt}/T) - \\ 1.3816 \times 10^{-7}\,(10^{11.344(1 - T/T_{bt})} - 1) + \\ 8.1328 \times 10^{-3}\,(10^{-3.49149(T_{bt}/T - 1)} - 1) + \log p_{bt} \tag{1}$$

where log refers to the logarithm in base 10, $p$ is the SVP, $T$ is the absolute atmosphere temperature in degrees Kelvin, $T_{bt}$ is

the steam-point temperature, and $p_{bt}$ is $p$ at the steam-point pressure. $T_{bt}$ is 373.15° K, $p_{bt}$ is 1013.25 hPa. The computing environments include homogeneous computing using only the Intel x86 CPU, homogeneous computing using only the MPE, heterogeneous computing using both CPUs and GPUs, and heterogeneous computing using both MPEs and CPEs. The FORTRAN codes of the Goff-Gratch equation are the same in all homogeneous computing environments (CPU-only and MPE-only), as shown in **Fig. 3a**. In all heterogeneous computing environments (CPU + GPU and MPE + CPE), the Goff-

Gratch equation is implemented for the GPU with Compute Unified Device Architecture (CUDA) FORTRAN (**Fig. 3b**) and the CPE in a hybrid mode where the MPE major task (in FORTRAN) manages CPE sub-tasks (in C-language) (**Fig. 3c**). Next, we give an example in software environment changes to measure the scale of the perturbations involved by slave cores. The Goff-Gratch equation in FORTRAN (**Fig. 3a**) is replaced by C-language using only the Intel x86 CPU. The input data $T$ is 234.917910298505. The floating-point results are shown in **Table 1**. For homogeneous computing, we can select the

combination of instruction sets and compilers to achieve BFB reproducible results. For heterogeneous computing, the fixed combination of instruction sets and compilers between master and slave cores generates inevitably a perturbation by slave cores. However, the perturbations involved by slave cores are not greater than such perturbations caused by software environment changes. From the discussions above, one key question to answer is, whether or not such perturbations caused by slave cores affect the scientific results? Next, we will design a methodology to characterize the uncertainties of

heterogeneous many-core architecture computing and discuss its influences on numerical solution.



# 3 Methodology to characterize uncertainties

## 3.1 The general idea to develop the methodology

As noted, heterogeneous many-core architecture computing can cause potential differences that lead to frequent ESM simulation perturbations by using GPUs or CPEs and generate unique uncertainties. However, identifying the computational perturbations and securing the model correctness with heterogeneous many-core architecture computing has two major challenges. First, the heavy legacy of codes limits the efficiency of refactoring and optimizing a complex ESM on heterogeneous systems, which makes it extremely difficult to examine the codes line by line. There exists an urgent demand on a straightforward metric to measure uncertainties instead of counting potential differences in each expression evaluation. Second, the complexity of the model makes it difficult for us to identify and mitigate the possible adverse impact of computational perturbations to the sciences enabled by the models. To overcome these challenges, our methodology includes: 1) designing a simple model that consists of typical discontinuous physical parameterizations defined by on-off switches to study the uncertainties due to the perturbations induced from slave cores or accelerators (the model should be simple enough so that porting, running, and result comparing between different supercomputing platforms can be easily performed); 2) developing an ensemble approach to characterize uncertainties quantitatively; 3) building up a software tool to verify the reliability of heterogeneous many-core systems.

## 3.2 A simple model to study potential differences

Intending to study uncertainties produced by potential differences, we design a multiple-column atmospheric model. First, to meet simplicity needs, the advection term describing the local variation due to its horizontal transport is used as a representation of the interaction between large scales and local convection (Li et al., 2016). Governing equations of the simple model are given by Eq.(2) and Eq.(3):

$$\frac{\partial T}{\partial t} = -u(z)\frac{\partial T}{\partial x} \qquad \text{and} \qquad (2)$$

$$\frac{\partial q}{\partial t} = -u(z)\frac{\partial q}{\partial x}, \qquad (3)$$

where $T$ and $q$ are the temperature and specific humidity, $u$ is the horizontal wind velocity as a function of height $z$. $u$ is fixed as time mean outputs from the climate simulation of CAM5 in the homogeneous multi-core platform at Qingdao Pilot National Laboratory for Marine Science and Technology (QNLM), during 1850-1860. The distance $\partial x$ is set to be about 277.5 km.

Second, the deep convective adjustment terms, $F_T$ and $F_q$, are used to control the water vapor content in the atmosphere (Emanuel and Živković-Rothman, 1999), which include on-off switch programming structure outputs, such as cloud bottom and cloud top. Governing equations become Eq.(4) and Eq.(5):

$$\frac{\partial T}{\partial t} = -u(z)\frac{\partial T}{\partial x} + F_T \qquad \text{and} \qquad (4)$$





$$\frac{\partial q}{\partial t} = -u(z)\frac{\partial q}{\partial x} + F_q ,$$

(5)

$F_T$ and $F_q$ are calculated in tendency equations using the ZM scheme (Zhang and Mcfarlane, 1995) along with the dilute convective available potential energy (CAPE) modification (Neale et al., 2008). With a set of thermodynamic properties of source air estimated from the grid-mean values at the level of maximum moist static energy zb and surface fluxes, a deep

convective updraft plume rises from zb with a specified lateral entrainment rate if the dilute CAPE is larger than 70 J kg-1 (Park et al., 2014). CAPE is defined by Eq.(6):

$$CAPE = \int_{Z_b}^{NBL} g \frac{T_{vp} - T_{ve}}{T_{ve}} dz ,$$

(6)

where $NBL$ is the neutral buoyancy level of the parcel lifted from the most unstable level in the boundary layer, $T_{vp}$ and $T_{ve}$ are the virtual temperatures of the parcel and environment, and $g$ is the acceleration due to gravity. $z_b$ is defined as the cloud

bottom. The cloud top $z_t$ is satisfied with Eq.(7):

$$[h_u(z_t) \leq h^*(z_t)] \& [h_u(z_t + 1) > h^*(z_t + 1)] = True ,$$

(7)

where $h_u$ is the moist static energy (MSE) of the lifted air parcel, $h^*$ is the saturation MSE of the environment (Wang and Zhang, 2018). MSE is defined by Eq.(8):

$$\frac{\partial m_u h_u}{\partial z} = E_u \bar{h} - D_u \hat{h} ,$$

(8)

where $m_u$ is the updraft cloud mass flux, $E_u$ and $D_u$ are the mass entrainment and detrainment rate, $\bar{h}$ and $\hat{h}$ are the MSE of grid-mean and detrained from updrafts.

Third, the vertical macrophysics adjustment terms, $Y_T$ and $Y_q$, are used to supplement large-scale stratiform precipitations. Governing equations become Eq.(9) and Eq.(10):

$$\frac{\partial T}{\partial t} = -u(z)\frac{\partial T}{\partial x} + F_T + Y_T$$

and  (9)

$$\frac{\partial q}{\partial t} = -u(z)\frac{\partial q}{\partial x} + F_q + Y_q ,$$

170  (10)

$Y_T$ and $Y_q$ are calculated in tendency equations using the Park stratus macrophysics scheme (Park et al., 2014). The Park scheme is defined as stratiform condensation/evaporation and cloud fraction parameterization (Donahue and Caldwell, 2018). The Park scheme diagnoses the liquid stratus fractions αl based on the assumption that the subgrid distribution of the total liquid relative humidity (RH) $v_l$ follows a triangular probability density function (PDF), where $v_l \equiv q_t / \bar{q_s}$, $q_t$ is the total liquid

specific humidity and $\bar{q_s}$ is grid-mean saturation specific humidity over water. The Park scheme also computes the grid-mean net condensation rate of water vapor into liquid stratus condensate.

The simple model is designed to simulate tropical areas where convection is most active. The model contains 144 columns in a latitude circle, with 1.9º × 2.5º resolution, a cyclic boundary condition, and 30 sigma vertical levels. The surface pressure is





fixed at 1000 hPa and the top model layer is about 2.26 hPa. The time integration step size is 30 min. The difference scheme

for advection is the Lax-Wendroff method. Initial conditions for T and q are obtained from CAM5 outputs on QNLM after

starting a spin-up of $10^5$ time steps.

We first give an example with homogeneous computing to study uncertainties of the simple model and illustrate the

influence of computational perturbations on simulation results. First, we design a mixed-language compiling mode of the

simple model in which the Goff-Gratch equation is replaced by C language. The FORTRAN and C version of the simple

model is with 64-bit variables and the same Intel compilers. Next, we change the variable precision in Goff-Gratch to 32-bit

to simulate larger perturbations. **Table 2** gives an example for the deviated digits of the mean surface air temperature (SAT)

as the model integration forwards in the simple model of Goff-Gratch equation in FORTRAN language compared to its C-

language version. **Figure 4** shows sudden changes in cloud bottom and top when the variable precision in Goff-Gratch is set

to 32-bit at 209 time steps. The results show that software changes can cause non-BFB reproducible results, and the

computational perturbations caused by the change in variable precision are large enough to cause obvious uncertainties.

For heterogeneous many-core architecture computing, compiler and instruction set differences can cause potential

differences as described in section 2.2. We use the simple model to study the uncertainties through the sudden changes in

cloud bottom and cloud top. We design seven simple model modes applied to homogeneous and heterogeneous computing as

listed in **Table 3**. The Intel mode is with homogeneous computing on a trusted machine. The simple model is implemented

for the GPU with CUDA FORTRAN and the CPE with hybrid schemes. PGI and MPE-only modes refer to compiling and

running with the same type of central processing units (CPUs) and MPEs, which is similar to homogeneous programs but

different from Intel mode in terms of compilers and processor architectures, while GPU-accelerated and CPE-parallelized

modes are heterogeneous programs. We take the result at 2.84° N latitude circle as an example to illustrate the outputs of the

simple model. For homogeneous computing, potential differences can not cause changes in cloud bottom and cloud top.

Heterogeneous many-core architecture computing can cause sudden changes compared with homogeneous computing at 255

time steps, as shown in **Fig. 5**.

### 3.3 An ensemble approach to characterize uncertainties

In this study, a quantitative analysis approach based on ensembles is used to characterize the uncertainties generated by

potential differences objectively. Characterizing the natural variability is difficult with a single run of the original simulation.

A large ensemble refers to a collection of multiple realizations of the same model simulation, generated to represent possible

system states. Ensembles created by small perturbations to the initial conditions are commonly used in climate modeling to

reduce the influence from the initial condition uncertainty and enhance model confidence (Sansom et al., 2013). We generate

a 100-member ensemble of 260 time steps in the simple model. The ensemble is formed by perturbing the initial temperature

with random noise multiplied by 0.1 from a Gaussian distribution of a zero mean and unit variance.

Statistical distributions collected from ensemble simulations help characterize the internal variability of the climate model

system (Baker et al., 2015). Note that measurements for characterizing uncertainties are estimates, where we ignore printing





phase errors of floating-point numbers (Andrysco et al., 2016). First, we compute the ensemble average of the mean horizontal standard deviation of the state variables to get a set of time series scores. Following **Table 3**, the Intel mode is with homogeneous computing on a trusted machine. The uncertainties are evaluated using root mean square error (RMSE) and mean absolute percentage error (MAPE) of scores between different modes. Next, for GPU-accelerated and CPE-parallelized modes, we add different magnitude order perturbations to function variables listed in **Table 4**, when transferred from GPUs to the CPU or CPEs to the MPE, in order to simulate accumulated potential differences by determining the critical state of the consistent climate.

### 3.4 A software tool to verify the reliability of heterogeneous many-core systems

We further discuss the software tool implementing the methodology to verify the reliability of heterogeneous many-core systems. Designing the simple model in homogeneous and heterogeneous modes is the basic work. In homogeneous modes, the simple model is the serial program operated by the FORTRAN language. Refactoring and porting the simple model in heterogeneous modes is the most demanding step. In this study, the simple model includes the dynamical process consisting of advection and physical parameterizations consisting of deep convective and macrophysics. To avoid data dependency, we only parallelize the parameterizations over different columns using GPUs or CPEs. The simple model codes on homogeneous and heterogeneous computing must be mathematically equivalent and stable.

Next, the time series of cloud bottom and top need to be compared to study uncertainties. Then, based on sudden changes of outputs, a 100-member ensemble of 260 time steps in the simple model is generated with different many-core architecture computing. Statistical distributions collected from ensemble simulations help characterize uncertainties including quantitative analyses of computational perturbations and acceptable error tolerances.

It is noted that, for a bounded model state variable (e.g., $q$), the probability often exhibits non-Gaussian distributions because of the lower bound. In this study, when $q$ falls below zero, it will be pulled back to zero (Li et al., 2016). In addition, we control some basic computing conditions, such as numerical stable codes like numeric constants written with the "d" notation (Bailey, 2008), no optimization, unified double precision variables, and 64-bit platform. Input files and ensemble simulation output files are in text format.

## 4 Experimental studies

### 4.1 The performance on the GPU-based HPC system

### 4.1.1 Brief description of the GPU-based HPC system at QNLM

The GPU computing system we used for our experiment consists of Nvidia Tesla V100. Each Tesla V100 GPU contains 80 multithreaded streaming multiprocessors (SMs) and 16 GB of global DDR4 memory. Each SM contains 64 FP32 cores, 32



FP64 cores, and 8 Tensor cores. ESMs are generally implemented for CUDA programs which are written to use massive numbers of threads, each with a unique index and executed in parallel (Kelly, 2010).

### 4.1.2 Results

Most physical parameterizations are structurally suitable for parallel architectures and demonstrate a high speedup when
migrating from CPU to GPU, such as the chemical kinetics modules (Linford et al., 2009) and the microphysics scheme (Mielikainen et al., 2013) in WRF (Weather Research and Forecasting model), the shortwave radiation parameterization of CAM5 (the 5th version of Community Atmospheric Model) (Kelly, 2010), and the microphysics module of Global/Regional Assimilation and Prediction System (GRAPES) (Xiao et al., 2013). Therefore, we implement the simple model on the GPU-based HPC system in QNLM.

Following sudden changes in cloud bottom and cloud top shown in **Fig. 5**, we discuss the influence of heterogeneous many-core architecture computing on the scientific correctness of numerical simulations on the GPU-based HPC systems. **Figure 6** shows the mean horizontal standard deviation of the simple model ensemble simulations in PGI and GPU-accelerated modes. The results show that heterogeneous many-core architecture computing also will not change the scientific correctness of simulation results on the GPU-based HPC system at QNLM.

We do quantitative analyses to characterize the uncertainties on the GPU-based HPC systems. We compute RMSE and MAPE among Intel, PGI, and GPU-accelerated modes, as shown in **Table 5**. The RMSE and MAPE of t between GPU-accelerated and PGI modes characterize uncertainties with heterogeneous many-core architecture computing with GPUs. The RMSE and MAPE between PGI and Intel modes characterize uncertainties due to software changes in homogeneous computing environments. The results of heterogeneous many-core architecture computing are larger than that of
homogeneous computing, which makes it easier to generate sudden changes of simulation results.

Next, we add $O(10^{-9}) \sim O(10^{-11})$ perturbations when GPUs transport data to the CPU, as described in Section 2.3. The PDFs of the simple model are shown in **Fig. 7** to ensure acceptable error tolerances when using GPUs. We find that the differences between PGI and GPU-accelerated modes are accompanied by the increasing magnitude order of perturbations.

### 4.2 The performance on Sunway TaihuLight system

#### 4.2.1 Brief description of Sunway TaihuLight

The Sunway TaihuLight is the first Chinese system to reach the number one of the Top500 list, which is built using Chinese homegrown heterogeneous many-core processors, SW26010. Its peak performance is 125 PFlops. Each SW26010 includes 4 MPEs, 256 CPEs, with the 64-bit instruction set and basic compiler components including C/C++, and FORTRAN compilers (Fu et al., 2016). Unlike the GPU-accelerated HPC systems where data transfer has to go between different processors and
accelerators, the on-chip heterogeneity of the SW26010 processor enables a uniform memory space to facilitate the data





transfer and leads to the uniform programming model between MPE and CPEs. ESMs are generally computed in a hybrid mode to use instruction sets separately where the MPE major task (in FORTRAN) manages CPE sub-tasks (in C-language).

### 4.2.2 Results

Considering great breakthroughs in optimizing high-resolution CESM on the heterogeneous many-core system (Zhang et al., 2020), we choose the Sunway TaihuLight as one of the heterogeneous running platforms. First, we focus on whether non-BFB reproducible results shown in **Fig. 5** generated by potential differences will cause scientific errors on the Sunway TaihuLight system. **Figure 8** shows the mean horizontal standard deviation of the simple model ensemble simulations in MPE-only_1 and CPE-parallelized_1 modes. The distribution is overall indistinguishable, which demonstrates that heterogeneous many-core architecture computing will not affect the scientific correctness of the simple model despite uncertainties on the Sunway TaihuLight system.

Although **Fig. 5** studies uncertainties in CPE-parallelized modes, it is necessary to do quantitative analyses to characterize uncertainties. We compute RMSE and MAPE among Intel, MPE-only_1, and CPE-parallelized_1 modes, as shown in **Table 5**. Next, we add $O(10^{-9}) \sim O(10^{-11})$ perturbations when CPEs transport data to the MPE on the Sunway TaihuLight system, as described in Section 2.3. The PDFs of the simple model are shown in **Fig. 9**. We find that with the increasing magnitude order of perturbations, the difference between MPE-only_1 and CPE-parallelized_1 modes with additional perturbation becomes larger. It is noted that potential differences should be in a certain range with the utilization of CPEs on the Sunway TaihuLight system.

### 4.3 The performance on the new Sunway system

The new Sunway system is built using an upgraded heterogeneous many-core processor SW26010P, which is similar to SW26010 in terms of architecture. ESMs are also generally computed in a hybrid mode.

As a new heterogeneous system, the new Sunway requires reliability verification to prepare for ESMs code porting. Following the sudden changes shown in **Fig. 5**, we discuss its influence on the scientific correctness of numerical simulation on the new Sunway system. **Fig. 10** shows the mean horizontal standard deviation of the simple model ensemble simulations in MPE-only_2 and CPE-parallelized_2 modes. The results show that heterogeneous many-core architecture computing also will not change the scientific correctness of simulations on the new Sunway system.

Quantitative analyses of uncertainties are also required on the new Sunway system. We compute RMSE and MAPE between MPE-only_2 and Intel and CPE-parallelized_2 and MPE-only_2, as shown in **Table 5**. Finally, we add $O(10^{-9}) \sim O(10^{-11})$ perturbations when CPEs transport data to the MPE on the new Sunway system. The PDFs of the simple model are shown in **Fig. 11** to ensure the acceptable error tolerances when using CPEs.





## 5 Summary and discussions


Numerical simulation advancements which demand tremendous computing power drive the progressive upgrade of modern supercomputers. In terms of architecture, due to physical and heat limits, most of the large systems in the last decade came in the heterogeneous structure to improve the performance continuously. Currently, heterogeneous many-core architectures include graphics processing units (GPU) accelerators and the Sunway hybrid structure consisting of master and slave cores.

There exist differences in compilers and instruction sets between master (CPU) and slave cores (accelerators) in heterogeneous many-core architecture computing environments. Therefore, compared with homogeneous CPU computing, heterogeneous numerical integration can cause perturbations in Earth system simulation and generate uncertainties whenever a slave core or accelerator is involved. Hence, characterizing uncertainties and comprehending whether it affects the scientific results of modeling in heterogeneous many-core architectures are urgently demanded.

In this study, we explore methodology to characterize the uncertainties of Earth system modeling with heterogeneous many-core architecture computing and understand the scientific consequence of perturbations caused by a slave core or accelerator. The developed method includes a simple multiple-column atmospheric model consisting of typical physical processes sensitive to perturbations, an efficient ensemble-based approach to characterize uncertainties. The simple model is used to study the perturbation-caused uncertainties through the sudden changes in cloud bottom and cloud top by applying to

homogeneous and heterogeneous systems that include GPU-based and Sunway HPC systems. First, in the homogeneous CPU computing environment, we add perturbations to simulate the heterogeneous behavior when slave cores involve the computation and examine the influence of perturbation amplitudes on the determination of cloud bottom and cloud top in both homogeneous and heterogeneous systems. Then, we compute the probability density function (PDF) of generated clouds in both homogeneous and heterogeneous computing environments with the increasing magnitude order of

perturbations. It is found that heterogeneous many-core architecture computing generates the consistent PDF structure with the one generated in homogeneous systems, although heterogeneous computing can slightly change the instant layer index of cloud bottom and cloud top with small perturbations within tiny precision differences. A series of comparisons on PDFs between homogeneous and heterogeneous systems show consistently acceptable error tolerances when using slave cores in heterogeneous many-core architecture computing environments.

Our current efforts demonstrate that perturbations involved by slave cores would not affect the scientific result of the simple model. However, refactoring and optimizing the legacy ESMs for new architectures requires verification in the form of quality assurance. The traditional tools, such as Community Earth System Model ensemble-based consistency test (CESM-ECT), focus on evaluating climate consistency within homogeneous multi-core architecture systems (Baker et al., 2015; Milroy et al., 2016). For heterogeneous many-core architecture computing, such tools cannot distinguish code errors from

unavoidable computational perturbations by slave cores or accelerators. Based on the CESM-ECT, we are going to develop a new tool to verify the correctness of ESMs on heterogeneous many-core systems in a follow-up study. Such a tool first shall include the ensemble that captures the natural variability in the modeled climate system in the mode of master core only and

perturbations in master-slave core parallelization. Second, the tool shall have a new function to measure the consist behavior of the ensemble as the perturbation magnitude increases. Eventually, the tool uses a quantitative criterion to measure the

correctness of ESMs on heterogeneous HPC.

Climate science advances and societal needs require higher and higher resolution Earth system modeling to better resolve regional changes/variations as well as extreme events. Given that the model resolution is intractable with computing resources available, higher and higher resolution Earth modeling demands greener supercomputing platforms with more affordable energy consumption. In the future, the heterogeneous hardware shall progressively have advances to achieve

better performance and lower energy consumption. Quality assurance of heterogeneous many-core computing environments is critical for building confidence in ESM porting, optimizing, and developing. Our methodology provides a protocol for verifying the reliability of new heterogeneous many-core systems.

**Code and data availability**

Codes, data and scripts used to run the models and produce the figures in this work are available on the Zenodo site

(https://doi.org/10.5281/zenodo.6481868, Yu et al., 2022) or by sending a written request to the corresponding author (Shaoqing Zhang, szhang@ouc.edu.cn).

**Author contributions**

Yangyang Yu is responsible for all plots, initial analysis and some writing; Shaoqing Zhang leads the project, organizes and refines the paper; Haohuan Fu, Lixin Wu and Dexun Chen provide significant discussions and inputs for the whole research;

all other co-authors make equal contributions by wording discussions, comments and reading proof.

**Competing interests**

The authors declare that they have no conflict of interest.

**Acknowledgments**

This research was supported by the National Natural Science Foundation of China (Grant No. 441830964) and Shandong

Province's "Taishan" Scientist Program (ts201712017) and Qingdao "Creative and Initiative" Frontier Scientist Program (19-3-2-7-zhc). All numerical experiments are performed on the homogeneous and heterogeneous supercomputing platforms at Qingdao Pilot National Laboratory for Marine Science and Technology and Wuxi National Supercomputing Center.



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





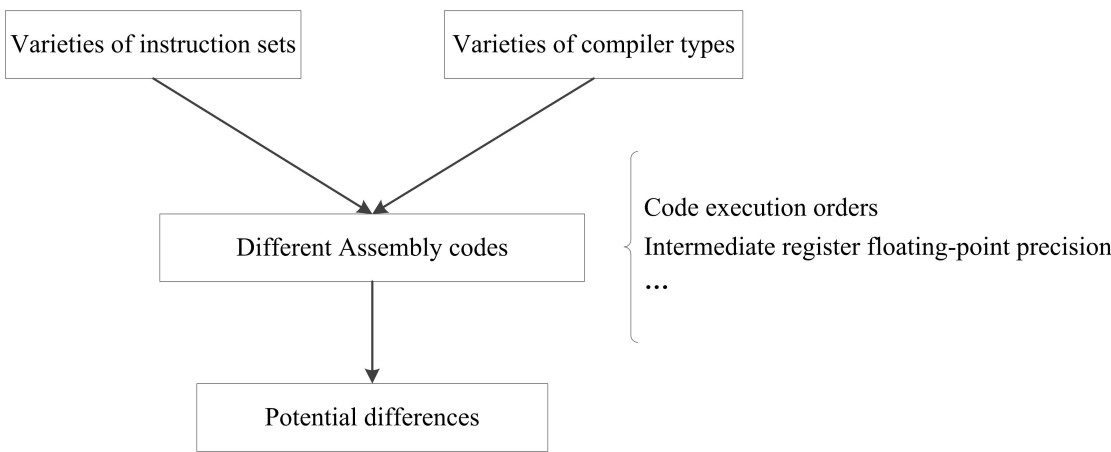

**Figure 1: Schematic illustration of sources that make potential differences in the result of floating-point computation.**

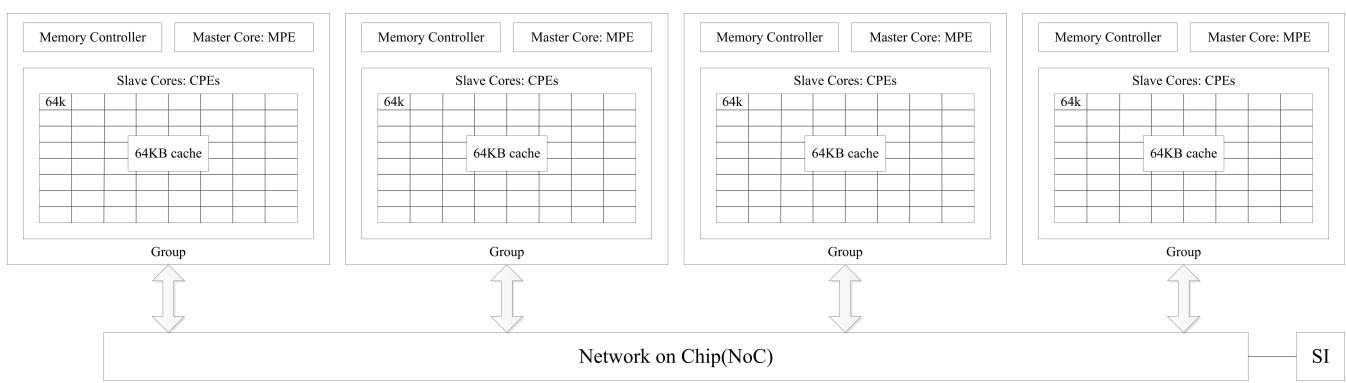

**Figure 2: A schematic illustration of the general architecture of the Sunway SW26010 processor. Each processor consists of 4 Core Groups, and each Core Group includes a Memory Controller, a Master Core (MPE) and 64 Slave Cores (CPEs), each of which has**
**a 64-KB scratchpad fast memory, called LDM (local data memory). 4 Core Groups are linked together by the Network on Chip, and the whole CPU is linked with other CPUs by the System Interface (SI) network (Courtesy to Fu et al., 2016).**





```
subroutine gofff(t,p,tbt)
    !parameter declaration
    real(8),intent(in) :: t, tbt
    real(8),intent(out) :: p

    !execution
    p = 10.0d0**(-7.90298d0*(tbt/t-1.0d0)+5.02808d0*
    log10(tbt/t)-1.3816d-7*(10.0d0**(11.344d0*(1.0d0-
    t/tbt))-1.0d0)+8.1328d-3*(10.0d0**(-3.49149d0*
    (tbt/t-1.0d0))-1.0d0)+log10(1013.25d0))
end subroutine                                    (a)
```

```
subroutine gofff(t,p,tbt)
    !parameter declaration
    real(8) :: t, p, tbt

    !local variable declaration
    real(8), device :: es_d

    !call gpu
    call goff_gpu <<<1,1>>>(t,tbt,es_d)
    p=es_d
end subroutine(b)

attributes(global) subroutine goff_gpu(t, tbt, p)
    !parameter declaration
    real(8), value, intent(in) :: t, tbt
    real(8), intent(out) :: p

    !execution
    p = 10.0d0**(-7.90298d0*(tbt/t-1.0d0)+5.02808d0*
    log10(tbt/t)-1.3816d-7*(10.0d0**(11.344d0*(1.0d0-
    t/tbt))-1.0d0)+8.1328d-3*(10.0d0**(-3.49149d0*
    (tboil/t-1.0d0))-1.0d0)+log10(1013.25d0))
end subroutine                                    (b)
```

```
subroutine gofff(t, p, tbt)
    !parameter declaration
    real(8) :: t, p, tbt

    !local variable declaration
    type para
        real(8) :: t, tbt
        integer(8) :: es
        end type
    real(8) :: es_array(1)
    para%es = loc(es_array(:))

    !call cpe
    call athread_init()
    call athread_spawn(slave_goff_parallel, para)
    call athread_join()
    p = es_array(1)
end subroutine

void slave_goff_parallel_(void *para){
    pe_get(para, &spara, sizeof(zm_convr_args_cc));
    t = spara.t;
    tbt = spara.tbt;
    slave_goff_(&t, &tbt, &svp);
    putmemreal(svp);         }

subroutine slave_goff(t,tbt,p)
    !parameter declaration
    real(8) :: t, p, tbt

    !execution
    p = 10.0d0**(-7.90298d0*(tbt/t-1.0d0)+5.02808d0*
    log10(tbt/t)-1.3816d-7*(10.0d0**(11.344d0*(1.0d0-
    t/tbt))-1.0d0)+8.1328d-3*(10.0d0**(-3.49149d0*(tbt/t-
    1.0d0))-1.0d0)+log10(1013.25d0))
end subroutine                                    (c)
```

**Figure 3:** The codes of the Goff-Gratch equation in homogeneous and heterogeneous computing environments.





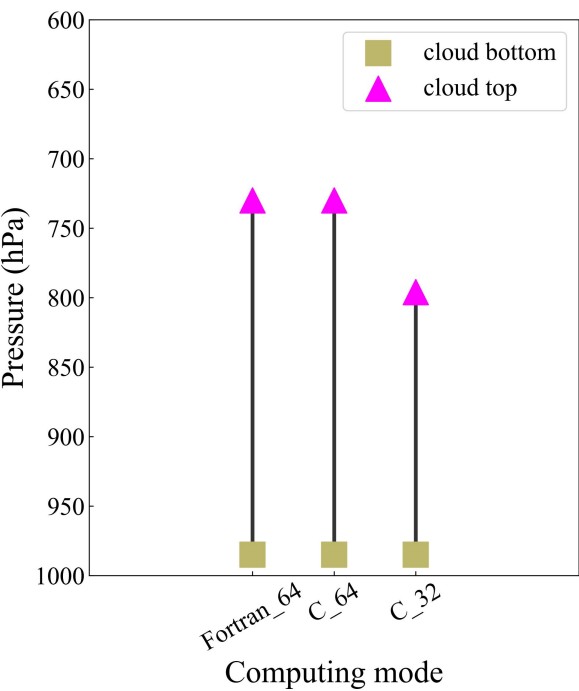

**Figure 4: The cloud bottom and cloud top in the simple model of Goff-Gratch equation in FORTRAN and C language at 209 time step.**

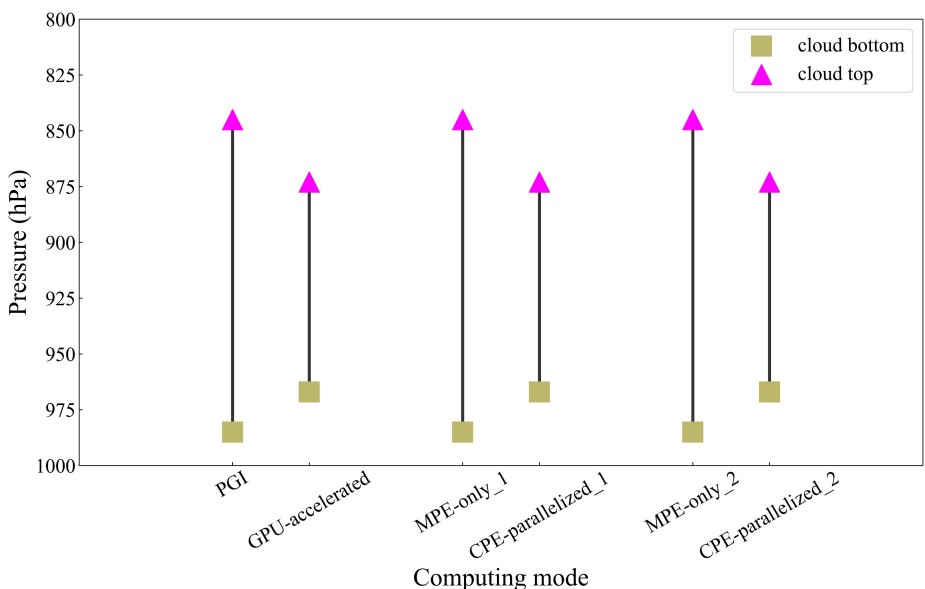

**Figure 5: The cloud bottom and cloud top with homogeneous computing and heterogeneous many-core computing on the GPU-based HPC system, Sunway TaihuLight, and the new Sunway at 255 time steps.**





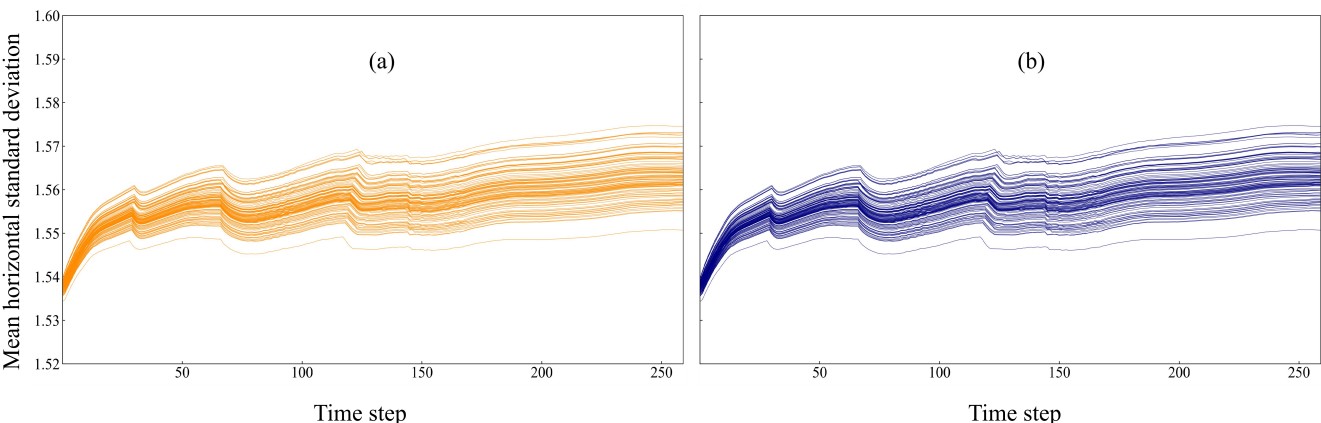

**Figure 6: The mean horizontal standard deviation of atmosphere temperature at 2.84° N latitude circle in (a) PGI and (b) GPU-accelerated modes.**

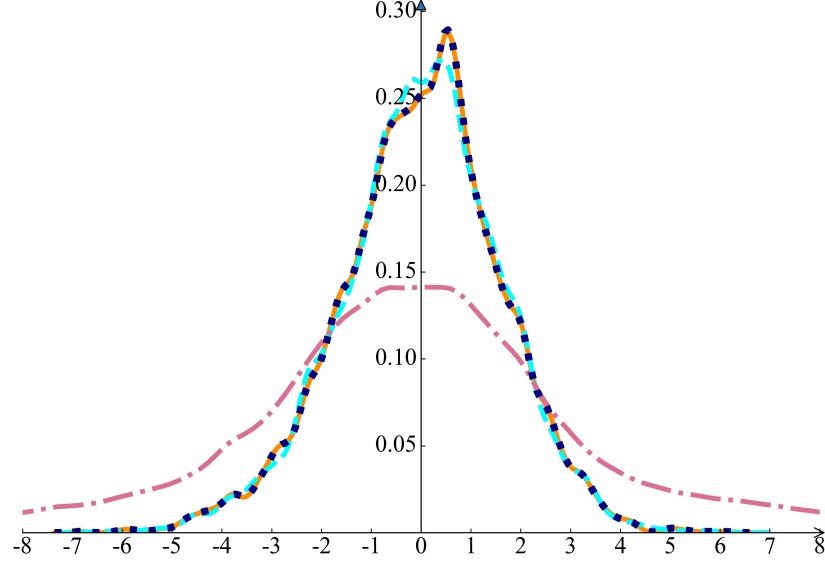

**Figure 7: The PDFs of atmosphere temperature of PGI and the GPU-accelerated modes with the increasing magnitude order of perturbations. The PDF of PGI is represented by the orange line. The PDFs of GPU-accelerated with the O(10⁻¹¹), O(10⁻¹⁰), and O(10⁻⁹) perturbations are represented by the black-dot, cyan, and pink lines. Note that the orange line and black-dot line are overlapped.**

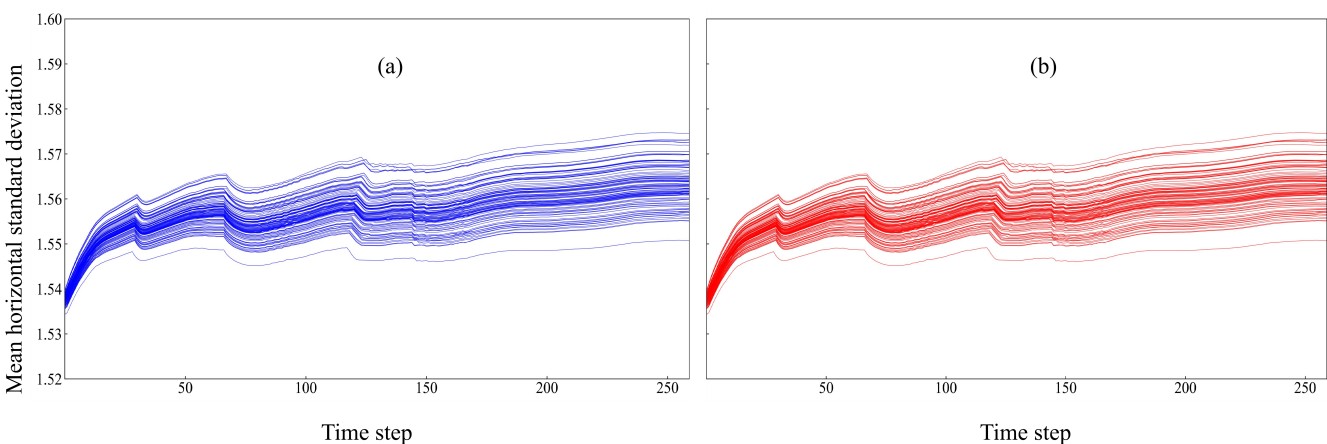

**Figure 8: The mean horizontal standard deviation of atmosphere temperature at 2.84º N latitude circle in (a) MPE-only_1 and (b) CPE-parallelized_1 modes.**

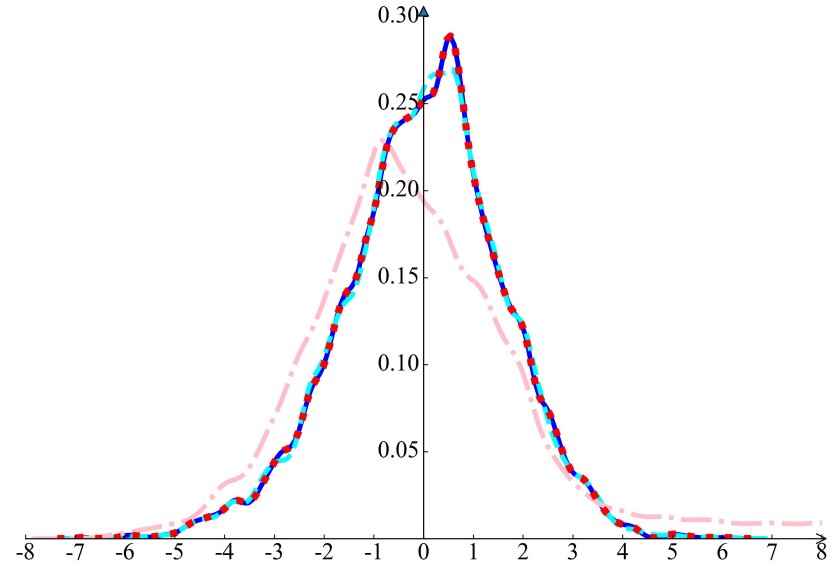

**Figure 9: The PDFs of atmosphere temperature of MPE-only_1 and the CPE-parallelized_1 modes with the increasing magnitude order of perturbations. The PDF of MPE-only_1 is represented by the blue line. The PDFs of CPE-parallelized_1 with the O(10⁻¹¹),**

**O(10⁻¹⁰), and O(10⁻⁹) perturbations are represented by the red-dot, cyan, and pink lines. Note that the blue line and red-dot line are overlapped.**



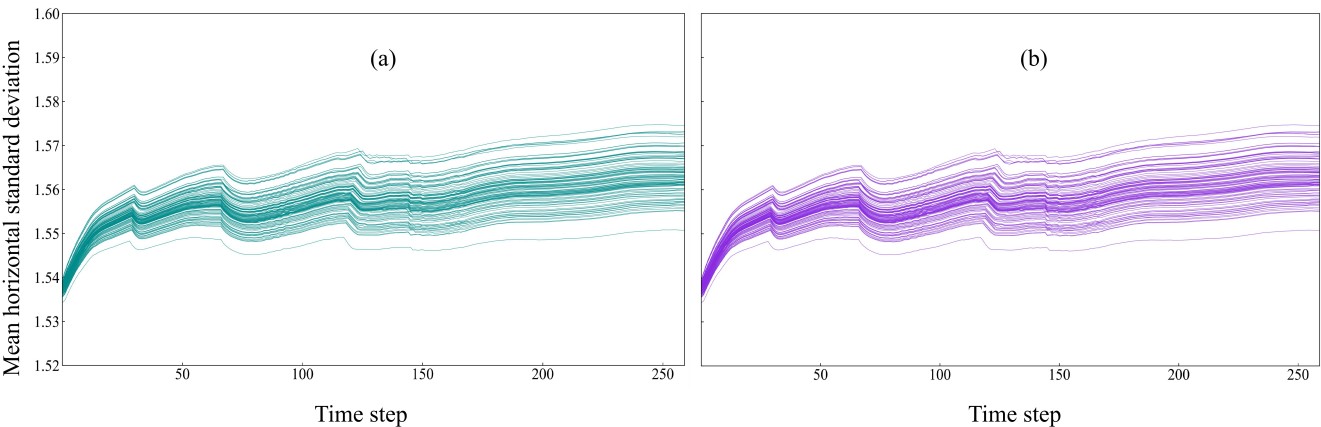

**Figure 10: The mean horizontal standard deviation of atmosphere temperature at 2.84° N latitude circle in (a) MPE-only_2 and (b) CPE-parallelized_2 modes.**


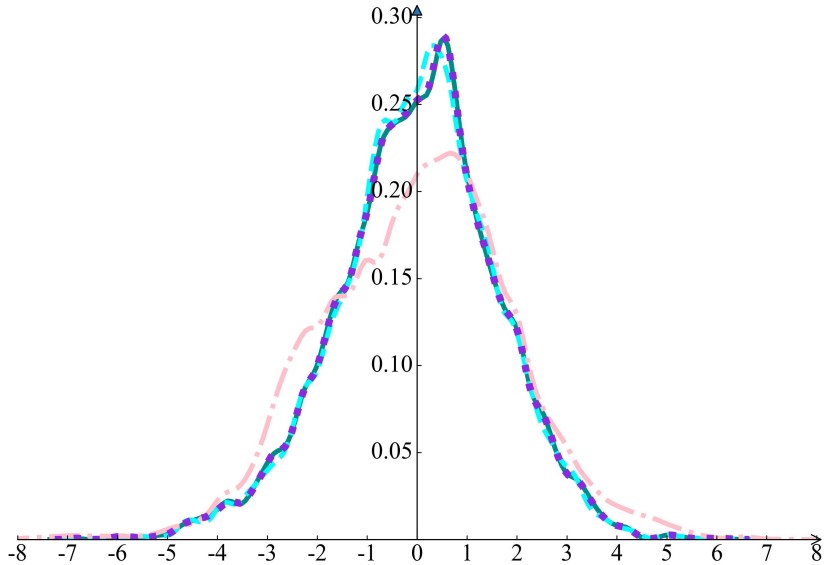

**Figure 11: The PDFs of atmosphere temperature of MPE-only_2 and the CPE-parallelized_2 modes with the increasing magnitude order of perturbations. The PDF of MPE-only_2 is represented by the darkcyan line. The PDFs of CPE-parallelized_2 with the $O(10^{-11})$, $O(10^{-10})$, and $O(10^{-9})$ perturbations are represented by the blueviolet-dot, cyan, and pink lines. Note that the darkcyan line**
**and blueviolet-dot line are overlapped.**





**Table 1: The results of the Goff-Gratch equation with homogeneous and heterogeneous computing in digits.**

| Computing environments | SVP (hPa) |
|---|---|
| CPU-only (FORTRAN-language) | 0.22678036581470054 |
| CPU-only (C-language) | 0.22678036581470031 |
| MPE-only | 0.22678036581470054 |
| CPU + GPU | 0.22678036581470056 |
| MPE + CPE | 0.22678036581470056 |

**Table 2: Movement of the difference of mean SAT at 2.84˙ N latitude circle in digits.**

| Time step | Modes | Mean SAT values |
|---|---|---|
| 207 | Intel_FORTRAN_64 | 297.3420582502919 |
| | Intel_C_64 | 297.3420582502854 |
| | Intel_C_32 | 297.3420594324838 |
| 208 | Intel_FORTRAN_64 | 297.3380180254846 |
| | Intel_C_64 | 297.3380180254779 |
| | Intel_C_32 | 297.3380192136227 |
| 209 | Intel_FORTRAN_64 | 297.3341208554172 |
| | Intel_C_64 | 297.3341208554104 |
| | Intel_C_32 | 297.3341221574373 |
| 210 | Intel_FORTRAN_64 | 297.3301317492619 |
| | Intel_C_64 | 297.3301317492551 |
| | Intel_C_32 | 297.3301324290993 |
| 211 | Intel_FORTRAN_64 | 297.3263169594554 |
| | Intel_C_64 | 297.3263169594486 |
| | Intel_C_32 | 297.3263172368548 |




**Table 3: The list of computing modes for the simple model**

| Modes | Compilers | Platforms |
|---|---|---|
| Intel | Intel 14.0.4 | Commercial supercomputing platform of Wuxi National Supercomputing Center |
| PGI | PGI 20.7 | Commercial GPU-based supercomputing platform of QNLM |
| GPU-accelerated | PGI 20.7 with CUDA FORTRAN | |
| MPE-only_1 | SW5 -host scheme | Sunway TaihuLight of Wuxi National Supercomputing Center |
| CPE-parallelized_1 | SW5 -master, -host and -hybrid schemes | |
| MPE-only_2 | SW9 -host scheme | New Sunway of QNLM |
| CPE-parallelized_2 | SW9 -master, -host and -hybrid schemes | |

**Table 4: The list of variables added perturbations**

| Variables | Descriptions | Subroutines |
|---|---|---|
| *qtnd* | Specific humidity tendency | ZM scheme |
| *heat* | Dry static energy tendency | ZM scheme |
| *s_tendout* | Dry static energy tendency | Park stratus macrophysics scheme |
| *qv_tendout* | Vapor specific humidity tendency | Park stratus macrophysics scheme |

**Table 5: The RMSE and MAPE of atmosphere temperature between different modes.**

| Modes | RMSE | MAPE |
|---|---|---|
| PGI - Intel | 7.850459998668024e-14 | 4.139396161561094e-12 |
| GPU-accelerated - PGI | 6.552539325839034e-07 | 3.765010369172171e-05 |
| MPE-only_1 - Intel | 7.002198572669633e-14 | 3.339540769718363e-12 |
| CPE-parallelized_1 - MPE-only_1 | 6.552540147495826e-07 | 3.765010851457739e-05 |
| MPE-only_2 - Intel | 5.508617595949462e-14 | 2.586305759079799e-12 |
| CPE-parallelized_2 - MPE-only_2 | 6.552540299899869e-07 | 3.765010862815439e-05 |