# Peer review of "Characterizing Uncertainties of Earth System Modeling with Heterogeneous Many-core Architecture Computing"

_Geoscientific Model Development, 2022_

## Author Response (AR1)

A point-by-point response to the review 1:

This paper presents a method for identifying and understanding the characteristics of computation uncertainties of Earth system modeling in heterogeneous architectures. The computation uncertainties and acceptable error tolerances in GPU-based high-performance computing and Sunway systems have been analyzed. The development of the Earth system models in heterogeneous systems is becoming more and more popular. This paper provides foundation support for developing and porting Earth system models in heterogeneous high-performance computing systems. The paper reads well and can be accepted after minor revision. See the below:

RE: Thanks for the reviewer's thorough examination of our manuscript (MS) and positive comments. We all agree that the comments are very constructive for us to improve the presentation of the MS, and all the major comments and other points have been fully addressed in the revision. Specifically, in the revision, we have added: 1) more detailed descriptions of the difference in codes between homogeneous and heterogeneous computing environments, 2) more detailed descriptions of the mixed-precision experiments, 3) more detailed descriptions of Table 3, etc.
The point-by-point replies are followed.

In the introduction, I am confused about the sentence "regular increases in the number of supercomputing processors came to a stop roughly one decade ago." However, the number of supercomputing processors has been increasing in the recent year. I suggest to change it to "regular increases in the processing frequency of supercomputing processors came to a stop roughly one decade ago."

RE: Thanks for the good suggestion. We have changed the sentence to "regular increases in the processing frequency of supercomputing processors came to a stop roughly one decade ago." Please see lines 32-34. Thanks.

Figure 3 shows a lot of similar codes. I hope the authors state clearly the difference in codes between homogeneous and heterogeneous computing environments.

RE: We have added the description of the difference in codes between homogeneous and heterogeneous computing environments. Please see lines 124-127. Thanks.

In Table 2, I am confused about the results that the mean SAT values are different between Intel_C_64 and Intel_C_32 modes in the tenth significant digit. However, the 32-bit variables have only 7 significant digits. Please state clearly the experimental results.

RE: In the mixed-precision experiments, only the variable precision in the Goff-Gratch function is changed to 32-bit. The main program is still with the 64-bit variables and is used for output. We have added the description of the mixed-precision experiments. Please see lines 207-213. Thanks.

Table 3 lists seven simple model modes applied to homogeneous and heterogeneous computing environments and the Intel mode is not shown in the later experiments. I hope the authors state clearly the role of the Intel mode.

RE: The Intel mode is with homogeneous computing on a trusted machine. The uncertainties are RMSE and MAPE of ensemble mean scores between different modes which are listed in Table 3. We have added the description of the Intel mode. Please see lines 242-244. Thanks.

I hope the authors point out the types and versions of all compilers in Section 3.3

RE: We have added the description of all compilers in Table 3 and cited Table 3 in Section 3.3. Please see lines 238-244. Thanks.

A point-by-point response to the review 2:

The paper is addressing a very important and underrepresented topic in scientific computing, namely a method to verify whether code which is ported to new hardware is still scientifically correct. The paper provides a new approach to tackle the problem using ensemble methods. The paper would still need to improve in clarity before being published in GMD following the comments below.

RE: Thanks for the reviewer's thorough examination of our manuscript (MS) and positive comments. We all agree that the comments are very constructive for us to improve the presentation of the MS, and all the comments have been fully addressed in the revision. Specifically, in the revision, we have added: 1) more detailed descriptions of uncertainties in heterogeneous many-core architecture computing, 2) more detailed descriptions of the simple model, 3) citation of references, etc.
The point-by-point replies are followed.

The paper would need to be place better into the existing literature. For example: The problems that are discussed in the paper (and in fact also the solution) have been discussed in publications on the use of reduced numerical precision. Here, ensemble methods are used to diagnose the impact of a precision reduction. See for example:
D. Dueben, A. Subramanian, A. Dawson and T. N. Palmer. A study of reduced precision to make superparametrisation more competitive using a hardware emulator in the OpenIFS model. Journal of Advances in Modeling Earth Systems, 9 (1), 566-584, 2017
Tintó Prims, O., Acosta, M. C., Moore, A. M., Castrillo, M., Serradell, K., Cortés, A., and Doblas-Reyes, F. J.: How to use mixed precision in ocean models: exploring a potential reduction of numerical precision in NEMO 4.0 and ROMS 3.6, Geosci. Model Dev., 12, 3135-3148, https://doi.org/10.5194/gmd-12-3135-2019, 2019.
The paper could also discuss the use of stochastic hardware, which would cause similar issues regarding checks of solution quality, see e.g.:

https://royalsocietypublishing.org/doi/full/10.1098/rsta.2013.0276

Finally, there is very interesting work by Oliver Fuhrer and team to port Earth system applications without changes of bit reproducibility, e.g.:

https://ieeexplore.ieee.org/document/6877351

RE: Thanks for the good suggestion. We have cited these references in the revision. Please see lines 205-206, 231-233, 34-35, 69-73. Thanks.

There is one point that I do not understand in the discussion about many-core architecture: You can already trigger non-identical results when running a serial CPU code with a changed compile flag. This is well known. Why are many-core architecture any different? Do they produce results which are different between different realisations of the same code on the same machine in the same setting? If no, I do not understand why many-core hardware is any different. If yes, you would need to provide more information how different different realisations actually are, for example in tables 1+2.

RE: Yes, we agree that homogeneous computing environments can already bring lots of different uncertainties, either due to hardware architecture transitions (such as the recent transitions from IBM processors to Intel processors around the year of 2010), or compiler configurations. The heterogeneous computing approach brings even more sources of uncertainties. Firstly, the heterogeneous way of computing brings an additional level of domain or task decomposition, when compared with the homogeneous way of computing. Therefore, the different algorithmic design would already bring different layouts of data elements and different sequence of computing. Secondly, we would, in some cases, face hardware difference between the general processor and the accelerator processor in a heterogeneous scenario. For example, in the SW26010 processor, the MPE core and the CPE core take a slightly different hardware design to implement floating-point arithmetic, thus leading to hardware-generated differences in corner cases related to denormalized numbers

(similar cases between CPU and early GPU architectures). Due to the above factors, a heterogeneous computing environment (CPU + GPU or MPE + CPE) can make non-identical results from a homogeneous computing environment (CPU-only or MPE-only). For the experiments, the same FORTRAN codes are used to implement the Goff-Gratch equation, and the same parameter and input data are used to run the program. Either CPU + GPU or MPE + CPE is on the same machine and forms a heterogeneous many-core architecture computing environment. The different realizations are defined as whether the Goff-Gratch equation is calculated in the slave core (GPU or CPE) or master core (CPU or MPE). The difference in computing environments between master and slave cores can make non-identical results whenever a CPE or GPU is involved. For example, the difference in math libraries of CPU and GPU can cause different floating-point results for a given input. We have added the detailed descriptions of the experiments in Section 2.2 and Tables 1,2 in the revision. Please see lines 78-88, 127-131, 133-138, and Tables 1,2. Thanks.

Also, optimised MPI parallelisation when using many CPUs in parallel can also show differences in results when running the same code in the same setting with the same number of nodes/cores as messages can arrive in different orders. How is this the problem that you find with master and slave architecture different? Does the difference really justify having a fully figure 2 on the Sunway architecture?

Re: Optimised MPI parallelisation can cause differences in results as messages can arrive in different orders. As mentioned in our reply to the above comment, in heterogeneous MPI computing, we face the same uncertainty from different MPI communication sequences. In addition, we also face uncertainty caused by further decomposition between the general core and the accelerator cores. Now, to discuss between a heterogeneous implementation on a CPU-GPU machine and a heterogeneous implementation on a Sunway MPE-CPE machine, the different granularity in the number of cores and the size of buffers would lead to different design strategies. For example, Fu et al., 2017 do adjustments of both the

computational sequence and the loop structures, so as to achieve a suitable level of parallelism for CPE clusters. The pure hardware setup of a Intel core, a GPU CUDA core, a Sunway MPE, and a Sunway CPE would also be different. Therefore, we think the heterogeneous challenge is different from the MPI random sequence problem. The Sunway situation would also be different from the CPU-GPU case. We have added the descriptions in the revision. Please see lines 78-88. Thanks.

LL126: It should also be mentioned that the models are simulating chaotic dynamics resulting in differences between simulations to grow exponentially. Porting linear models would be simple.

RE: Thanks for the good suggestion. We have added the description of the simple model in the revision. Please see lines 156-157. Thanks.

LL131: I understand the reason why you are applying the method to a small model in the paper. However, you should outline somewhere how you would apply your approach in a model as large as an Earth system model. Also, I do not understand what you mean by "building a software tool". You are presenting a method applied to a customised code. What is the tool you are talking about?

RE: Thanks for the good suggestions. We have added the description of applying the approach in a model as large as an Earth system model in the revision. Please see lines 365-369. Then, we have changed the statement of "building a software tool" to "implementing an application" in the revision. Please see lines 157-158.

Figure 6,8,10: The caption indicates that you are showing standard deviations, but the Figures seem to show ensemble members. I am confused as standard deviations should be a single number. Or is this a variable of the model?

RE: In Figures 6,8,10, we computed the mean horizontal standard deviation of

atmosphere temperature for each ensemble member at a latitude circle to get a set of time series scores. We have added the description of the computing method of the mean horizontal standard deviation in the revision. Please see lines 239-241.

The English should be improved if possible.

RE: Thank you for your valuable and thoughtful comments. We have carefully checked and improved the English writing in the revised manuscript.

Minor comments:

L19: Not really "potential differences". The model will most of the time not be bit reproducible when the hardware is changed and therefore different.

RE: Thanks for your advice. We have modified the sentences, please see line 19. Thanks.

L24-25: I do not understand what is meant by "on-off switches" or why this is useful.

RE: "on-off switches" is the selection programming structure. We have modified the description, please see lines 23-25. Thanks.

L32: The development does not require an increase in computing power. The increase in resolution does.

RE: Thanks for your advice. We have modified the sentences, please see line 32.

L116: "inevitable a perturbation" Is this reproducible between different runs or stochastic?

RE: The perturbations are reproducible between different runs. The master cores and

the slave cores take a slightly different hardware design to implement floating-point arithmetic, thus leading to hardware-generated differences in corner cases related to denormalized numbers. Compared with homogeneous computing using the master cores only, heterogeneous computing can cause nonidentical floating-point outputs whenever a slave core or accelerator is involved. We have modified the sentence, please see lines 78-88. Thanks.

L145: I guess this should be a "\Delta x"?

RE: Yes. We have modified the sentence, please see line 169. Thanks.

L215: "of different modes" What is meant by this? Is the error calculated between the Intel mode and other modes when using the same random perturbation for each ensemble member?

RE: The uncertainties are evaluated between different modes which are listed in Table 3. We have modified the sentences, please see lines 240-242. Thanks.

L221: "the basic work" I do not understand.

RE: In this paper, designing the simple model in homogeneous and heterogeneous modes is the top priority. We have modified this sentences, please see line 250. Thanks.

"magnitude order perturbations" should be "perturbations of different order of magnitude" in several sentences throughout the paper

RE: Thanks for the good suggestions. We have modified the sentences throughout the paper. Thanks.

L249: What is QNLM?

RE: We have added the description of "QNLM". Please see lines 167-169. Thanks.